# Algorithm-aided engineering of aliphatic halogenase WelO5* for the asymmetric late-stage functionalization of soraphens

Johannes Büchler [1,2], Sumire Honda Malca[1], David Patsch[1,3], Moritz Voss[1], Nicholas J. Turner [2], Uwe T. Bornscheuer [3], Oliver Allemann[4,5], Camille Le Chapelain [4], Alexandre Lumbroso[4], Olivier Loiseleur[4✉] & Rebecca Buller [1✉]

Late-stage functionalization of natural products offers an elegant route to create novel entities in a relevant biological target space. In this context, enzymes capable of halogenating $sp^3$ carbons with high stereo- and regiocontrol under benign conditions have attracted particular attention. Enabled by a combination of smart library design and machine learning, we engineer the iron/α-ketoglutarate dependent halogenase WelO5* for the late-stage functionalization of the complex and chemically difficult to derivatize macrolides soraphen A and C, potent anti-fungal agents. While the wild type enzyme WelO5* does not accept the macrolide substrates, our engineering strategy leads to active halogenase variants and improves upon their apparent $k_{cat}$ and total turnover number by more than 90-fold and 300-fold, respectively. Notably, our machine-learning guided engineering approach is capable of predicting more active variants and allows us to switch the regio-selectivity of the halogenases facilitating the targeted analysis of the derivatized macrolides' structure-function activity in biological assays.

[1] Competence Center for Biocatalysis, Institute of Chemistry and Biotechnology, Zurich University of Applied Sciences, Einsiedlerstrasse 31, 8820 Wädenswil, Switzerland. [2] School of Chemistry, The University of Manchester, Manchester Institute of Biotechnology, Manchester M1 7DN, United Kingdom. [3] Institute of Biochemistry, Dept. of Biotechnology & Enzyme Catalysis, Greifswald University, Felix-Hausdorff-Strasse 4, 17487 Greifswald, Germany. [4] Syngenta Crop Protection AG, Schaffhauserstrasse 101, 4332 Stein, Switzerland. [5] Present address: Idorsia Pharmaceuticals Ltd, Hegenheimermattweg 91, 4123 Allschwil, Switzerland. ✉email: olivier.loiseleur@syngenta.com; rebecca.buller@zhaw.ch

Subtle molecular changes in small molecules can have a profound impact on their biological activity and metabolism. For example, monodechlorinated and didechlorinated vancomycin lose approximately 30 and 50% of the antimicrobial effect exhibited by parent antibiotic vancomycin[1], respectively. Similarly, the introduction of a single methyl group led to MK-8133, a dual orexin receptor antagonist, with 480-fold boosted potency[2]. In the latter example, the methyl group had to be installed through a laborious five-step *de novo* synthesis[3]. In contrast, late-stage functionalization (LSF) of C–H bonds offers direct access to new analogs of a lead structure. In this way, LSF constitutes a valuable tool to investigate structure-activity relationships of small molecules, especially natural products[4], and supports the optimization of on-target potency, selectivity, and absorption-distribution-metabolism-excretion (ADME) properties while helping to improve physical properties such as solubility and stability. In addition, LSF can be of aid in the protection and exploration of novel intellectual property space by giving access to molecular entities left unexplored by conventional synthetic approaches[3]. Typical functionalizations of C–H bonds include oxygenation, amination, methylation, borylation, thiantrenation, azidation, and halogenation[3,5]. Notably, the incorporation of chlorine and bromine offers new routes to modify the molecule through cross-coupling chemistry or substitution reactions[6].

The synthesis route to organohalides commonly involves multiple steps. In order to achieve high chemo-, regio- and stereoselectivities[7], the use of protecting, directing, or activating groups is often necessary. As some of these groups may need to be removed in subsequent steps, such approaches lack atom economy. Overall, the halogenation of unactivated C–H bonds remains a challenge for chemists[8,9]. Enzymatic halogenations, on the other hand, often exhibit excellent regio- and stereoselectivity even in complex molecular settings, therefore complementing— and sometimes outperforming— existing strategies[10–13].

Biocatalytic halogenations are carried out by enzymes called halogenases, which are typically classified according to their catalytic mechanism: Heme, vanadium, and flavin-dependent halogenases (Fl-Hals) follow an electrophilic aromatic substitution mechanism via the generation of hypohalous acid, iron/α-ketoglutarate dependent halogenases (αKGHs) employ a radical pathway, while *S*-adenosyl-L-methionine (SAM) fluorinases react via a nucleophilic substitution[14]. In contrast to the electrophilic Fl-Hals, which act on electron-rich $sp^2$-carbons through the intermittent generation of hypohalous acid, αKGHs can functionalize unactivated $C(sp^3)$-H bonds. The catalytic mechanism is based on the generation of a high-valent $Fe^{IV}=O$ intermediate capable of abstracting a hydrogen atom from the substrate. The resulting carbon radical is then coupled to the iron-coordinated chlorine, thereby affording the corresponding halogenated compound in a regio- and stereoselective manner (Fig. 1a). In recent years, a handful of αKGHs have been described: The carrier-protein dependent halogenases BarB1 and BarB2[15], SyrB2[16], CytC3[17], CmaB[18], HctB[19], CurA[20] and the synthetically more interesting freestanding halogenases WelO5[21], WelO5*[22], *Wi*-WelO15[23], AmbO5[24], the BesD[25] family, the recently identified plant halogenases SaDAH and McDAH[26] as well as the halogenase AdeV[27], which acts on nucleotide substrates.

To date, halogenase engineering has mainly focused on Fl-Hals[10,28–33] or haloperoxidases[34,35] with the aim to provide catalysts capable to derivatize non-natural substrates en route to more valuable aryl-, alkoxy or amino acid compounds[36–40] or for their use as final products[41,42]. In contrast to the wealth of reports on Fl-Hals, the number of αKG-dependent halogenases is small and their reported substrate scope is mainly limited to their natural substrates and close analogs. In 2019, the first examples of engineering freestanding αKGHs toward non-natural substrates were reported by us and others[23,43]. The studies highlighted the malleability of αKGHs WelO5* and *Wi*-WelO15 by tailoring the enzymes for the regio- and stereoselective chlorination of a non-alkaloid type substrate and more closely related substrate analogs of 12-*epi*-hapalindole C, respectively. In both cases, substantial increases in apparent $k_{cat}$ (WelO5*: 400-fold compared to wild type; *Wi*-WelO15: 276-fold compared to first-generation mutant) could be achieved by enzyme engineering[23,43]. Despite the pioneering nature of these engineering studies, it should be noted, that the chosen non-natural substrates were similar in size and shape to the halogenases' natural substrate 12-*epi*-hapalindole C.

Soraphens are the largest known family of myxobacterial polyketides and display a diverse array of chemical moieties (e.g., unsubstituted phenyls and sensitive allylic ethers amongst other features) which render them an attractive test case for an application to a broader range of polyketides. Soraphen A, the main representative of the soraphens, was identified in the supernatant of the *Sorangium cellulosum* strain Soce26 and shows inhibitory activity against several phytopathogenic fungi through inhibition of acetyl-coenzyme A carboxylase (ACC)[44]. The crystal structure of the yeast biotin carboxylase (BC) domain complexed with soraphen A (PDB ID: 1W96) revealed that the macrolide acts as an allosteric inhibitor[45] by disrupting dimerization of the BC domain and stabilizing the catalytically inactive monomer (Supplementary Fig. 1)[46]. Even though highly potent, the further development of these natural products as potent antifungal agents has been hampered due to off-target selectivity concerns and sensitization in mammals[47]. Notably, soraphen A has recently also become a target of pharmaceutical interest[44]. In cancer therapy research, several studies established that tumoral cells have a dependence on *de novo* fatty acid synthesis and that inhibition of ACC triggers apoptosis with no or little effects on healthy cells[48]. Modified lead structures are therefore sought after, both in agrochemistry as well as in pharmaceutical chemistry, which—owing to the complexity and sensitivity of the natural product—are, however, difficult to obtain in the quantities and within the timeframes required by modern drug discovery[49]. Consequently, the development of adapted synthetic methodologies, including biocatalytic transformations, are of key interest to drive the development of complex, natural compounds into useful products.

In this work, we assess the biocatalytic potential of αKGHs by employing algorithm-assisted enzyme engineering to tailor the recently described non-heme iron halogenase WelO5* from *Hapalosiphon welwitschii* IC-52-3 for selective halogenation of soraphen A (**1**), soraphen C (**2**) and their semi-synthetic analogs **3** and **4** (Fig. 1b). Phenotypic testing of the derivatized macrolides against six different fungal key pathogens in crop protection is carried out to inform about the halogenated macrolides' biological activity.

## Results

**Synthesis of starting material.** Soraphen structures contain ten stereocenters, including hydroxyl-, methyl, methoxy, and a hemiacetal group rendering these natural products biologically highly interesting but chemically very complex molecules. In addition, such polyketide macrocycles are also known to adopt several conformations[50]. While soraphen A can be accessed through an optimized bioprocess[47,51], its penultimate biosynthetic congener soraphen C is a much less explored member of the soraphen family and very difficult to isolate in sufficient amounts from fermentation despite its value as a chemical probe[52]. To obtain the compound for our study, we, therefore, developed a concise semisynthesis starting from soraphen A

**Fig. 1 Proposed reaction mechanism and substrates of wild type and engineered WelO5* variants. a** Proposed reaction mechanism of Fe(II)/αKG-dependent halogenases and hydroxylases. Mechanism adapted from Mitchell et al.[66] and Galonic et al.[79] **b** Structural comparison of the macrolide soraphen A and its analogs (**1-4**) with WelO5*'s natural substrate 12-epi-fischerindole U (**5**)[22] and the accepted martinelline-derived fragment (**6**)[43].

(Supplementary Fig. 2). This route, entailing selective oxidative demethylation of the allylic methoxy group and a subsequent stereo-directed reduction of the intermediate ketone, offers the first synthetic access to soraphen C. Even though soraphen C had been obtained earlier through fermentation[53], we are now reporting the first complete characterization of this natural product.

**Identification of an active starting halogenase for halogenation of soraphen A.** To identify a halogenase which would accept soraphen A, an enzyme panel consisting of 59 native and engineered electrophilic and freestanding aliphatic halogenases capable of acting on a wide range of $sp^2$ and $sp^3$-carbons was screened (Supplementary Tables 1, 2). The engineered Fl-Hals included in the panel were derived from literature[31,41,42,54] whereas the engineered αKGHs consist of WelO5* variants which we had previously identified as possessing a broadened substrate scope[43].

All halogenases, expressed in *E. coli* BL21(DE3), were used for crude cell-lysate biotransformations of soraphen A in a deep-well plate. While neither halogenation nor hydroxylation activity toward the target substrate was detected for any of the wild type enzymes, liquid-chromatography coupled to mass spectrometry (LC-MS) analysis showed that biotransformations with 26 out of the 28 included WelO5* variants led to the formation of derivatized soraphen A. In particular, variants V81G/I161P, V81G/I161G, as well as I161A, showed appreciable activity leading to the detection of three prominent products with $m/z$ ratios of 577.2 and 559.2, which are consistent with the calculated mass of two chlorinated products and a hydroxylated product, respectively (Supplementary Fig. 3). The structures of the chlorinated products **1a** and **1b**, as well as the hydroxylated product **1c**, were solved using nuclear magnetic resonance (NMR) analysis, which confirmed chlorination and hydroxylation of aliphatic carbon centers of **1** (Supplementary Fig. 4). Notably, the enzymatic derivatization occurred at positions in the molecule which would have been difficult to target via traditional chemical means and opens options for further functionalization in previously unexplored segments of the molecule.

In contrast to previous engineering studies on WelO5*, the reaction selectivity (halogenation vs. hydroxylation) of the best-performing variant V81G/I161P was slightly in favor of the halogenation reaction (2:1 halogenation to hydroxylation ratio,

estimated via the SIM areas of the product peaks). This is remarkable for the transformation of a structurally highly divergent compound compared to the natural substrate 12-epi-fischerindole U (Fig. 1b). As has been observed for WelO5* and other αKGHs, this enzyme family's reaction selectivity is strongly governed by the substrate structure and substrate positioning in the active site. It has been shown that the biotransformation of 12-epi-hapalindole C, another literature-known native substrate of WelO5* similar in structure to 12-epi-fischerindole U[22], led to the predominant formation (ca. 50%) of hydroxylated product and 25% of the desired chlorinated product 12-epi-hapalindole E[43]. Other examples include studies on the carrier-protein-dependent halogenase SyrB2, which turned into an effective hydroxylase in response to the length of added C-atoms in its native substrate L-threonine[55].

**Enzyme engineering of WelO5* for improved activity and selectivity.** While wild type WelO5* did not accept soraphen A, mutation of only two residues near the active site conferred initial halogenation and hydroxylation activity toward the bulky macrolide substrate. This activity data underlines the striking malleability of WelO5*[23,43] allowing a considerable expansion of substrate scope by exchange of very few amino acids strategically positioned in the vicinity of the reactive iron species. Docking of soraphen A into a model of the best-performing WelO5* variant V81G/I161P, which was created using SWISS-MODEL[56], led to solutions in which the active site was capable to accommodate soraphen A (Fig. 2). Based on these docking results and in agreement with the studies from Hayashi et al.[43] and Duewel et al.[23], three critical amino acid positions, namely 81, 88, and 161 (Fig. 2), were chosen for full randomization in a library targeted for the use in an algorithm-aided enzyme engineering strategy.

Traditionally, gene mutagenesis methods for the generation of variant libraries are PCR-based techniques and include error-prone polymerase chain reaction (epPCR), saturation mutagenesis, or DNA shuffling. Saturation mutagenesis, as required in our approach, is a highly advantageous technique in rational enzyme design, however, it is known to suffer from amino acid bias leading to reduced library quality and thus increased screening effort[57]. In order to allow for an unbiased library construction, we opted for a *de novo* library synthesis using high-fidelity on-chip solid-phase gene synthesis[58]. This library construction strategy allowed us to limit library diversity to the theoretical 8000

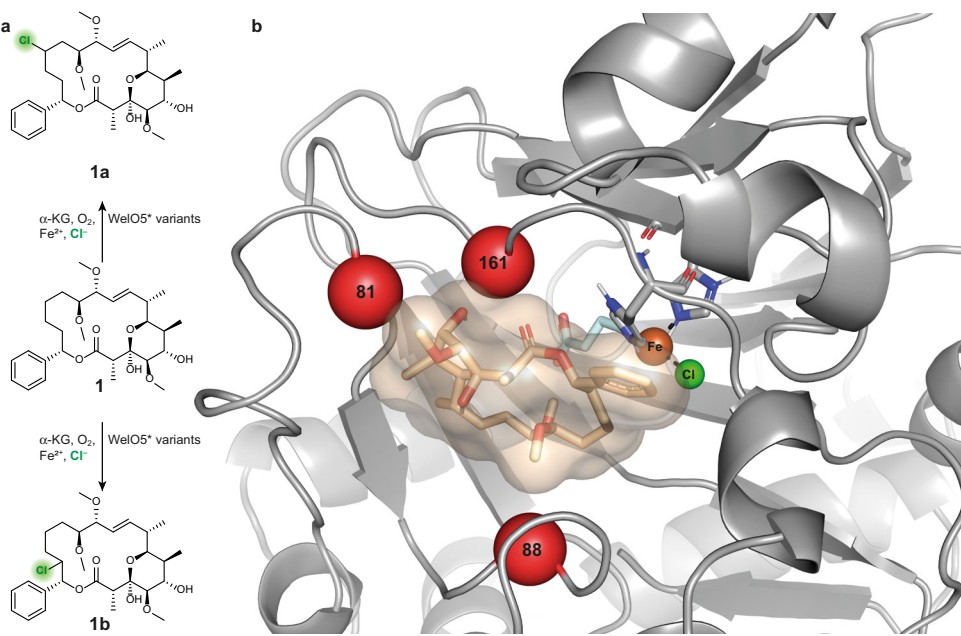

**Fig. 2 Identification of target sites for enzyme engineering. a** Regio-divergent halogenation of soraphen A in function of the employed WelO5* variant. **b** Docking of soraphen A (wheat) into a model of variant WelO5* V81G/I161P (gray). The enzyme model was created using SWISS-MODEL[56] and the crystal structure of WelO5 (PDB ID: 5J4R) as a template. Soraphen A was docked using AutoDock Vina[77]. The red spheres indicate the targeted positions for the full randomization library. Histidine residues and the α-ketoglutarate (pale cyan) in complex with the iron (orange sphere) are shown as sticks. The chlorine coordinating to the iron is shown as a green sphere.

variants (20³) for the full co-randomization of residues at positions 81, 88, and 161 and minimize screening effort. For simplicity, we will report WelO5* variants with a three-letter code hereafter. For instance, wild type WelO5*, which contains the amino acids V81/A88/I161, is denoted as variant VAI, whereas variant V81G/A88A/I161P, which was identified as being active on soraphen A in the initial hit panel screening, is dubbed GAP.

The synthetic gene library was ordered from Twist Bioscience. The gene fragments were cloned into the pET28b(+) vector and transformed into *E. coli* BL21(DE3) cells in house. About 504 unique variants (6.3% of the theoretical library) were confirmed by Sanger sequencing and screened for the derivatization of soraphen A (Fig. 3, red circles). As expected, we observed the formation of the previously identified products in addition to a second hydroxylated compound (**1d**). Overall, four distinct soraphen A analogs could be produced by the analyzed enzyme variants: Chlorination products **1a** and **1b**, as well as hydroxylation products **1c** and **1d**, were observed (Supplementary Fig. 4). In all cases, hydroxylation product **1d** was a side product and formed only in minimal amounts (max. formation of 2%, not isolated).

In comparison to the previously best-performing variant GAP, we identified amino acid combinations (VIG, AVP, and TIA) that boosted total chlorination activity for soraphen A by 8-10-fold, whereas variant SLP increased the total halogenation activity by 13-fold. In addition to improving total chlorination activity, the three-site combinatorial library also contained variants, which modulated the regioselectivity of the halogenation reaction. Instead of preferentially forming product **1a**, variant LHS exclusively led to chlorination product **1b** while remaining similar in total chlorination activity to variant GAP.

While the theoretical number of unique variants in a *de novo* synthesized three-site combinatorial library is 8000, a much higher number of samples will have to be screened in practice. This is because the degree of oversampling increases with the

percentage of targeted library coverage. As a result, a library coverage of 95% will require the analysis of ~24,000 variants[59], an effort which demands considerable resources. Inspired by previous successful applications of machine learning in protein engineering[60–62], we explored the remaining protein landscape in silico using Gaussian processes, allowing us to reduce the physical screening burden and accelerate the accumulation of beneficial mutations. By representing amino acids as a 17-dimensional vector, which was obtained by concatenating the five-dimensional T-scale descriptor[63] and additional amino acid characteristics[64], our machine learning approach then defined the feature vector of a sequence by joining the vector representation of its individual amino acids at sites V81X, A88X, and I161X. With this strategy, we were able to identify both more active and more selective variants with noticeable accuracy and precision (Supplementary Fig. 5 and Supplementary Table 3). All seven variants predicted towards activity (Fig. 3, blue circles) were highly active, with four of them outperforming the previous best variant SLP (up to a 16-fold increase over GAP). Predictions toward selectivity (Fig. 3, green circles) show a similarly high fraction of improved variants, with one of them enhancing activity over the previously most selective variant for chlorination site B by >2-fold while retaining a complete selectivity for regioisomer **1b**.

While the detailed mechanism behind the improved activity of the evolved variants remains unclear, we attempted to get a better understanding of the factors governing the regioselectivity of the evolved variants by carrying out docking studies with substrate **1**. For these experiments, we used the available crystal structure of WelO5 (PDB ID: 5J4R), a close homolog of WelO5*, as a basis of our homology modeling with the tool SWISS-MODEL[56]. Comparing the docking results of variant GAP, our most selective variant for the production of **1a**, with the analysis of variant AHG, our most selective variant for the synthesis of **1b**, we observed a shift in substrate positioning with respect to the iron-oxo and the Cl-ligand (Fig. 4 and Supplementary Fig. 6). The set of mutations acquired in AHG presumably changes the binding

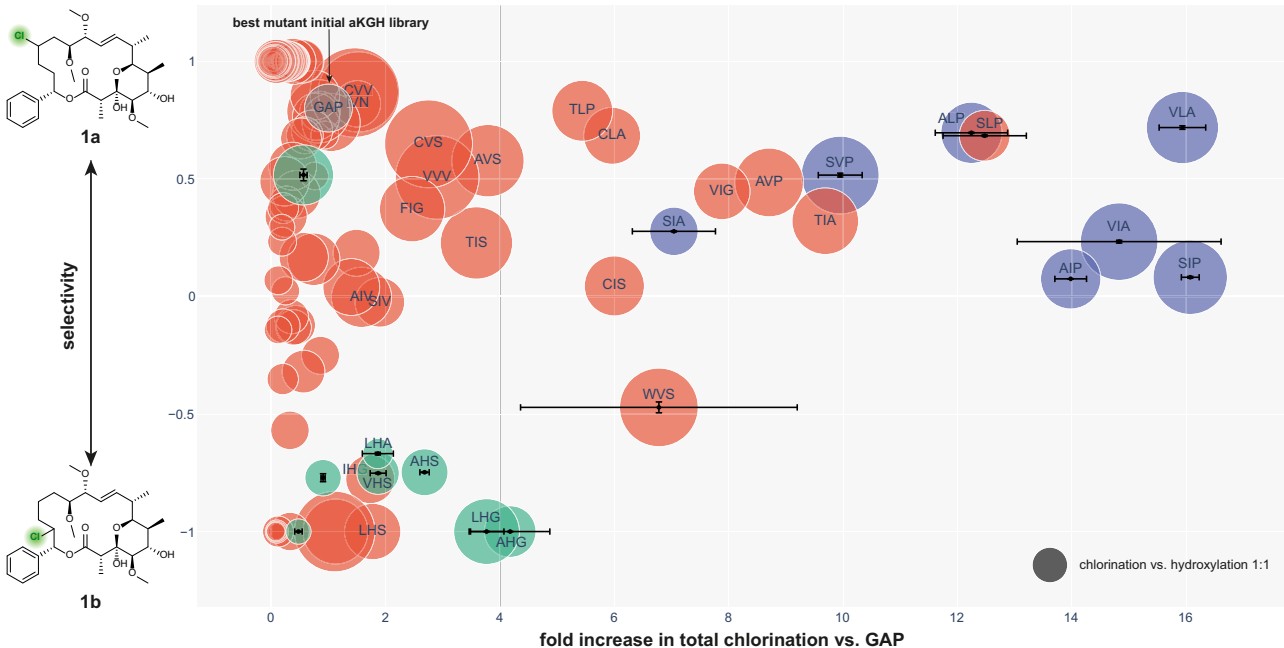

**Fig. 3 Biotransformation results of the combinatorial library of WelO5* (red) and the predicted variants (blue and green).** Two chlorinated products (**1a**, **1b**) of soraphen A were detected. The y-axis shows the regioselectivity of the chlorination. The selectivity (S) is calculated using the formula $S = (SIM_{1a} - SIM_{1b})/(SIM_{1a} + SIM_{1b})$. For variants showing a higher than 1.5-fold increase in total chlorination compared to WelO5* V81G/A88A/I161P (GAP, gray) the amino acid sequence of the three engineered positions is shown. On each measured 96-well plate, the variant GAP and negative controls were included as internal references. The combinatorial library variants were measured once. Predicted variants (selectivity; activity) and best-performing variants (SLP; WVS) were measured in triplicate as individual experiments. Data were presented as mean values ± standard deviation. The size of the circle corresponds to higher chlorination vs hydroxylation activity. As a reference, the dark gray sphere corresponds to a 1:1 halogenation to hydroxylation activity (area halogenation products/area hydroxylation products).

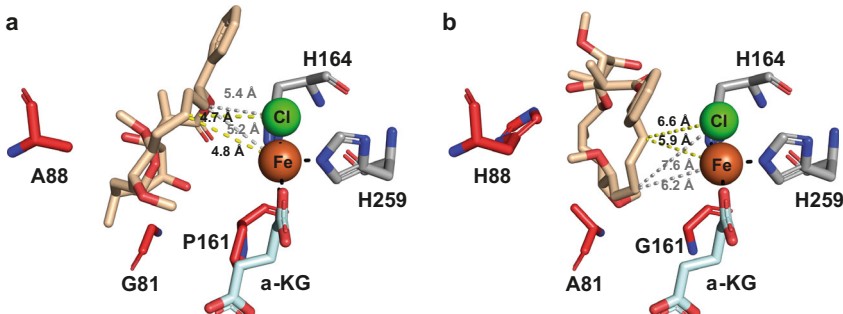

**Fig. 4 Soraphen A (wheat) docked models of the regio-divergent WelO5* variants GAP and AHG. a** In the model of WelO5* variant GAP (amino acids G, A, and P shown as red sticks) shorter distances between the iron and chloride to C14 of soraphen A (yellow dotted lines) than to C16 of the macrolide (gray dotted lines) suggest the structural reason for the predominant formation of regioisomer **1a**. **b** In the soraphen A docked model of WelO5* variant AHG (amino acids A, H, and G shown as red sticks), a shift in substrate positioning leads to shorter distances between the iron and chloride to C16 of soraphen A (yellow dotted lines) than to C14 of the macrolide (gray dotted lines) underlining the observation of selectivity for formation of regioisomer **1b**. Histidine residues (gray) and the α-ketoglutarate (pale cyan) in complex with the iron (orange sphere) are shown as sticks. The chlorine coordinating to the iron is shown as a green sphere.

mode of soraphen A in such a way, that H-abstraction is now favored from a different C–H bond, namely C16, instead of C14 as observed for GAP (Supplementary Fig. 7).

To further assess the substrate promiscuity of the engineered WelO5* variants and to expand our palette of uniquely derived soraphen analogs for biological testing, we analysed the transformation of soraphen C (**2**) and the soraphen analogs **3** and **4**. In analogy to soraphen A, we observed the formation of two chlorinated and hydroxylated products for soraphen C. Also, the soraphen analogs **3** and **4** led to the formation of several singularly derivatized macrolide structures (Supplementary Table 4). Interestingly, the initial whole-cell screening using

soraphen A as a substrate did not reveal doubly chlorinated or doubly hydroxylated products nor a mixture thereof. To further investigate the substrate promiscuity of our engineered variants, we continued by carrying out in vitro studies applying optimized reaction conditions using mono-chlorinated **1a**, **1b**, and **2a** as substrates and purified enzyme preparations of variants GAP, SLP, VLA, and WVS. Of all combinations tested, variants WelO5* SLP and VLA exhibited detectable substrate promiscuity and proved capable to produce minor amounts of doubly chlorinated products starting from **1a** (0.04% conversion with SLP) and **1b** (1.7% with SLP and 1.9% conversion with VLA) as well as a hydroxylated product derived from **1b** (3.7% with SLP

**Table 1 Biochemical characterization of selected WelO5* variants for the biocatalytic production of 1a.**

| Variant | app. $k_{cat}$ (min$^{-1}$) | app. $K_m$ (mM) | app. $k_{cat}/K_m$ (min$^{-1}$ mM$^{-1}$) | rel. $k_{cat}$ | TTN[+] |
|---------|------------------|-------------|--------------------------------|-----------|--------|
| GAP | 0.026 ± 0.007 | 0.45 ± 0.14 | 0.07 ± 0.21 | 1 | 0.3 ± 0.2 |
| SLP | 2.413 ± 0.349 | 0.42 ± 0.09 | 5.74 ± 0.07 | 93 | 30.0 ± 8.3 |
| VLA | 1.959 ± 0.509 | 0.44 ± 0.03 | 4.45 ± 0.07 | 75 | 91.8 ± 22.0 |

[+](TTN experiments were performed in two test series (biological replicates) and each series consisted of four independent experiments ($N = 4$); kinetic parameters are given as the average of $N = 3$ ± SD).

and 5.8% conversion with VLA) (Supplementary Fig. 8). Overall, and in alignment with the observations made for the halogenation of a martinelline-derived fragment by Hayashi et al.[43], the main detectable products of the engineered WelO5* variants under standard reaction conditions were the mono-derivatized soraphens.

**Biochemical characterization of improved WelO5* variants.** Following our enzyme engineering campaign, we explored the biochemical characteristics of our evolved halogenase variants. For variant GAP, our best initial hit, as well as for variants SLP and VLA, the most active variants for the biocatalytic production of **1a**, Michaelis–Menten kinetics were recorded (Table 1). As initial velocities decreased for all variants with increasing substrate load, a substrate inhibition model was used (Supplementary Eq. 1) to determine the kinetic parameters. Substrate inhibition is a common phenomenon in enzymology and is well documented for enzymes following a radical reaction mechanism. P450 enzymes, for example, have been shown to suffer from decreased activity at high substrate concentrations in function of the provided substrate[65]. Similarly, WelO5* kinetics seems to be governed by the substrate type: While non-classical Michaelis–Menten kinetics were observed when the engineered WelO5* variants were presented with the macrolide soraphen A, the martinelline-derived fragment **6** used in a previous study[43] did not elicit observable substrate inhibition in closely related WelO5* enzyme variants even at concentrations as high as 2.0 mM.

Analysis of the kinetic parameters revealed that variant VLA (apparent $k_{cat} = 1.96$ min$^{-1}$; TTN = 91.8) exhibited a > 75-fold improved apparent $k_{cat}$ and a > 300-fold increased total turnover number (TTN) yielding substantially improved concentrations of product **1b** (Supplementary Fig. 9) when compared to the initial hit, variant GAP (apparent $k_{cat} = 0.03$ min$^{-1}$; TTN = 0.3). Strikingly, engineered VLA displays a similar apparent $k_{cat}$ and total turnover number for the bulky macrolide soraphen A as wild type halogenases acting on their native substrates[12]: wild type WelO5, for example, is reported to halogenate its native substrate 12-*epi*-fischerindole U with a $k_{cat}$ of 1.8 min$^{-1}$ whereas the total turnover number is reported to be 70[24].

It was previously shown that WelO5*[43] and other αKGHs of bacterial[25,66,67] and plant origin[26], can install alternative anions. We, therefore, tested the ability of our best WelO5* variants (GAP, SLP, and WVS) to generate additional soraphen A derivatives using a panel of alternative anions, namely F⁻, Br⁻, I⁻, N₃⁻, and NO₂⁻. Among the anion tested, Br⁻, N₃⁻, and NO₂⁻ were incorporated into the substrate as shown by the appearance of up to two products with the expected $m/z$ ratios in selected ion monitoring (Supplementary Fig. 10), in analogy to the product pattern in the corresponding chlorination reactions. Incubation with iodide and fluoride under standard reaction conditions did not yield derivatized product likely due to steric and electronic reasons. As previously observed for WelO5* variants[43] and the freestanding plant halogenase SaDAH[26], the chloride and azide anion yielded the best transformation results

as deduced from SIM peak areas. The regioselectivity of alternative anion incorporation was not determined directly. Interestingly, however, distribution between the two observed products when incubating halogenases SLP and WVS with alternative anions reflected the observed product distribution in chlorination reactions leading us to postulate installation of bromide, azide, and nitrate at the same sites in the substrate molecule.

**Biological activity of soraphen derivatives against phytopathogenic fungi.** Next, we embarked on the biological characterization of the halogenated products. Toward this goal, the biotransformations of soraphen A and soraphen C were carried out at preparative scale (100 mg scale) using the optimized WelO5* variants VLA (soraphen A, halogenation product **1a**), WVS (soraphen A, halogenation product **1b**), and VAA (soraphen C, halogenation product **2a**). In all cases, enough product was obtained and submitted to biological activity profiling. The performed biological tests were phenotypic, i.e., carried out on living fungi, either with a fungal liquid culture or as a preventative application on leaf disk, and considered not only on-target potency but also metabolism, physicochemical properties (leaf penetration for instance), UV stability, and phytotoxicity. The activity is reported as BP80 (break point 80%), which corresponds to the concentration above which 80% of activity, measured as fungal growth inhibition, is observed (Fig. 5, Methods in SI). Six different fungi were evaluated (Fig. 5), as they represent key pathogens in crop protection and cause a large spectrum of crop diseases: *Puccinia recondita* (black rust), *Septoria tritici* (leaf blotch), *Erysiphe graminis* (also called *Blumeria graminis*, powdery mildew), and *Monographella nivalis* (snow mold) attack cereals, especially wheat, while *Botrytis cinerea* (gray mold) acts on horticultural crops including wine grapes, and *Mycosphaerella arachidis* (leaf spots) affects peanut plants. Finding natural molecules to fight these plant pathogens is of special relevance for Europe, where the European Green Deal[68] has become a driver for use of natural products in crop protection.

The aliphatic region of soraphen A, which was derivatized in our experiments, is known to make hydrophobic contact with the acetyl-coenzyme A carboxylase BC domain (in particular with W487, using numbering from PDB ID: 1W96). This critical tryptophan residue is highly conserved within the acetyl-coenzyme A carboxylase BC domain across the tested fungal species. Therefore, the conformational changes in the soraphens, which the chlorine or hydroxyl-group introduction was expected to induce, may also have resulted in a binding penalty which could have led to the observed reduced activity, specifically in the case of hydroxylated compound **1c**. Remarkably, though, all chlorinated analogs conserved a good level of activity on most fungal pathogens, which is unprecedented to date in the ensemble of derivatives accessible from the fully functionalized natural product[47].

As the biological tests performed were phenotypic, a target-based SAR analysis cannot fully explain the activity observed in vivo, which depends on many other factors such as in planta

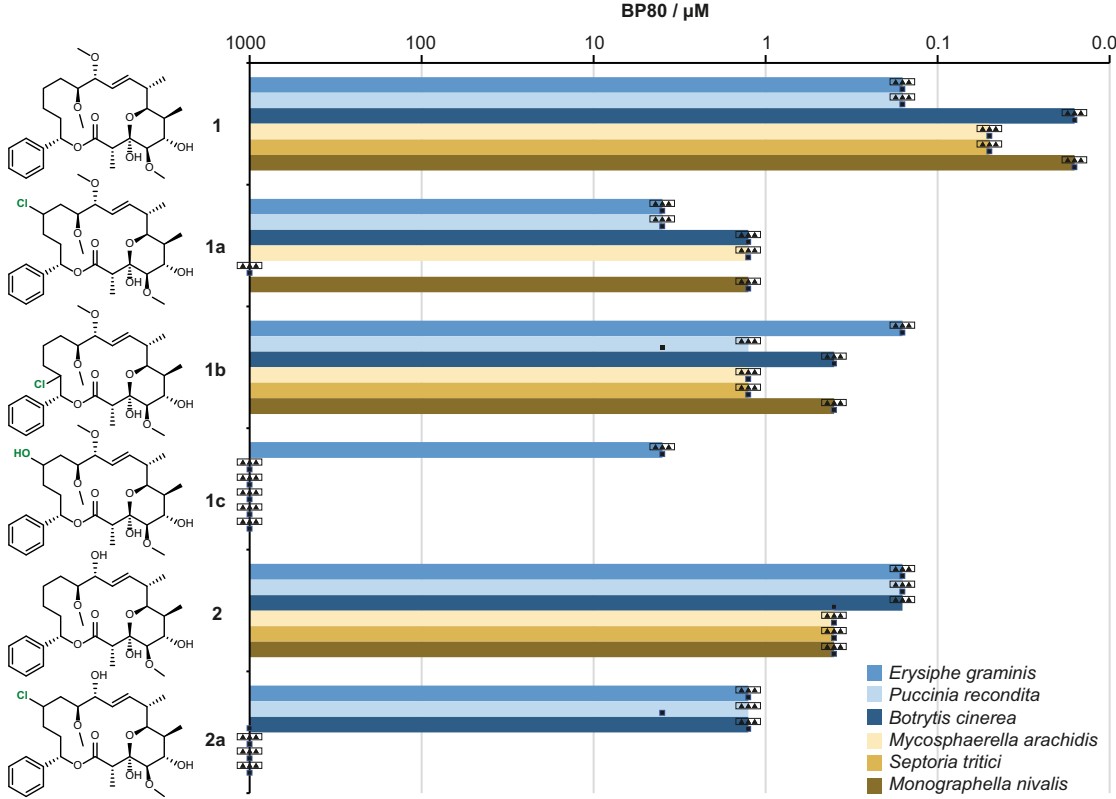

**Fig. 5 Break point of efficiency 80% (BP80).** Soraphen A (**1**), C (**2**), and enzymatically derivatized analogs (**1a**, **1b**, **1c**, **2a**) were tested against six different fungi, determined by dilution series at four concentrations and measured against positive and negative standards. BP80 represents the concentration of active ingredients above which 80% or more of efficiency is observed. The experiments on living organisms were performed in two test series (biological replicates). The first series consisted of a single experiment (N = 1, square) for each fungus, the second series consisted of triplicates with three samples tested in parallel during the same test session (N = 3, triangles).

and in fungi metabolism, cell penetration, distinct physicochemical properties of the compounds as well as on differential metabolism and even variations in plant-fungi interactions. Nevertheless, it is worth noting that the site of chlorination seems to impact observed biological activity, **1b** showing an overall better performance than **1a** whereas the chlorinated soraphen C derivative **2a** seems to display higher species selectivity than the other investigated compounds.

Altogether, the observed modulation of the soraphens' biological activity highlights the value of the enzymatic late-stage functionalization approach to generate knowledge in regions of the natural product structure very difficult to access by any chemical means. In fact, spanning over more than 30 years, comprehensive derivatization efforts on the soraphens, which aimed to evaluate whether modified structures might retain good bioactivity, failed: Even minor structural changes led to complete loss of potency[47]. In this context, the activity observed for the here reported chlorinated soraphen analogs and the relatively short time, in which they were obtained especially when compared to total or semisynthesis approaches is even more remarkable. These results represent a good starting point for further structure-activity studies of this class of macrolides and underline the ability of engineered WelO5* halogenases to display unique distance and geometry-based control of functionalization in complex molecules.

## Discussion

Here, we demonstrate that through the application of algorithm-assisted enzyme evolution, we endowed WelO5* variants with the capability to halogenate the bulky non-natural substrate soraphen A.

Our most active engineered variant WelO5* VLA catalyzes the halogenation of the macrolide **1** to yield product **1a** with an apparent $k_{cat}$ value and a total turnover number which mirror the activity of wild type aliphatic halogenases for their natural substrate (*vide infra*)[12] thus highlighting the malleability of WelO5*'s active site and underlining the effectiveness of our engineering strategy.

Following the identification of hot spots through rational enzyme design, the use of machine learning enabled us to successfully navigate the sequence-function space of a $20^3$ combinatorial library of aliphatic halogenase WelO5*. By providing a homogenous and consistent data set of high quality for training and validation of the algorithms, we were able to reliably predict functional properties such as activity and regioselectivity of the enzyme variants from sampling only 6% of the theoretical data points. To date, there are only a few examples that showcase the use of machine learning to improve an enzyme's activity[69,70], and the extent of sampling to obtain predictions varies strongly (Supplementary Table 5). To mature the field, further experimentally confirmed examples such as this one will be necessary to develop more standardized guidelines for the use of machine learning in enzyme engineering and enable comparison between predictors[69]. In addition, the implementation of molecular dynamics simulations into the enzyme engineering workflow might help to further fine-tune machine learning algorithms and —as automation hardware and library design strategies are similarly maturing—allow to interrogate sequence space even more effectively.

Through our resource-saving evolution process, we generated halogenase variants capable of functionalizing soraphen A and soraphen C yielding three distinct halogenated species in

quantities sufficient for biological testing. Notably, the enzymatically derivatized positions would have been difficult to target using organic chemistry methods, thus highlighting the potential of employing aliphatic halogenases for the late-stage functionalization of complex natural products. These structurally unique and selectively active natural products are desirable targets as they have already demonstrated their extraordinary power as shuttles to new biological target spaces[71–74].

Future efforts to understand the underlying structural factors to selectively derivatize non-native substrates will help to generalize evolution strategies for this enzyme family and algorithm-driven engineering as well as homology model-based docking approaches will play an important role in accelerating this process. Looking forward, aliphatic halogenases are rapidly becoming an interesting new tool for the development of biologically active molecules to be used, for example, in medicinal and agrochemistry.

## Methods

**Materials**. All chemicals and solvents were purchased from commercial suppliers (Sigma Aldrich, VWR, and Carl Roth) and were used without further purification. Phusion High-Fidelity DNA polymerase, T4 DNA ligase, and all restriction enzymes used in this study were purchased from New England BioLabs (Massachusetts, USA). Gene synthesis was performed by Twist Bioscience (California, USA). Oligonucleotides and sequencing service was provided by Microsynth AG (Balgach, Switzerland).

**Initial halogenase panel and protein expression**. Genes encoding halogenases in pET28b(+) were purchased from Twist Bioscience. Each plasmid was transformed into *E.coli* BL21(DE3) and the cells were plated on an LB agar plate containing 50 μg/mL kanamycin. A single colony of freshly transformed cells was cultured overnight in 1 mL of LB medium containing 50 μg/mL kanamycin. About 0.1 mL of the culture was used to inoculate 0.9 mL of TB medium supplemented with 50 μg/mL kanamycin and 0.2 mM IPTG (for Fl-Hal) or of 0.9 mL Zymo5052 auto-induction medium[75] supplemented with 50 μg/mL kanamycin (for αKGH) in a 96-well deep-well plate. Expression was carried out for 24 h at 20 °C, 300 rpm (5-cm shaking diameter) using a Duetz system (Kühner AG, Basel, Switzerland). The cells were pelleted by centrifugation at $4000 \times g$, 4 °C, for 15 min, and the supernatant was discarded. The cell pellet was stored in a −80 °C freezer prior to biotransformation reactions.

**Combinatorial library WelO5\***. WelO5* was subjected to simultaneous saturation mutagenesis of the three hot spots Val81, Ala88, and Ile161, leading to a theoretical library size of $20^3 = 8000$ mutants. The variants were obtained as a pooled gene fragment library from Twist Bioscience (California, USA) and subcloned a His-tag into a modified pET28b(+) expression vector, in which the nucleotide sequence between the NcoI and NdeI restriction sites was removed and the NcoI replaced by the NdeI restriction site. Consequently, inserting the gene with a terminal stop codon between the NdeI and XhoI restriction sites yields an ORF without His-tag. The cloning was realized with the In-Fusion HD Cloning Plus kit (Takara Bio, Shiga, Japan). The library was amplified with forward primer 5′-AAGGAGATATACATATGTCGAACAACACCATCTCGAC-3′ and reverse primer 5′-GGTGGTGGTGCTCGAGTTAGCTCCAATAGTAGATTTTGTTG-3′ using the DNA polymerase and a standard PCR protocol provided by the kit manufacturer. The gel-purified PCR product (NucleoSpin Gel and PCR Clean-up, Macherey-Nagel, Düren, Germany) was inserted into NdeI/XhoI-linearized pET28b(+) vector (modified) using the In-Fusion enzyme mix. The resulting reaction mixture was utilized to transform competent *E. coli* Stellar™ cells from the kit. After reconstitution in 1 mL SOC medium, 20–50 μL were spread on an LB kanamycin agar plate for transformant count and the remaining cell solution was inoculated into 50 mL LB kanamycin overnight growth at 37 °C. Plasmid isolated from 10 mL culture was used to transform competent *E. coli* BL21(DE3) cells. Clones from LB kanamycin agar plates were sampled for colony PCR to verify the presence of insert prior to sequencing. More than 1000 colonies were picked and grown separately in 96-deep-well plates for DNA Sanger sequencing (Microsynth AG, Balgach, Switzerland). For screening, strains containing empty vector, wild type WelO5*, and other WelO5* variants (positive controls) were included on each plate.

**Biotransformation αKGH**. The cell pellets were subjected to chemical lysis using 100 μL of 50 mM sodium phosphate buffer (pH 8.0) supplemented with 1 mg/mL lysozyme, 0.5 mg/mL polymyxin B, and 0.01 mg/mL DNase. Incubation was carried out for a minimum of 30 min at 20 °C on a shaking incubator at 850 rpm. Biotransformations were initiated by the addition of 100 μL of sodium phosphate buffer (pH 8.0) containing 2 mM substrate, 220 mM α-ketoglutaric acid sodium

salt, 212 mM sodium ascorbate, 1000 mM NaCl, and 2.6 mM ammonium iron(II) sulfate to each well. Assay plates were sealed with breathable membranes and incubated overnight at 20 °C on a shaking incubator at 850 rpm. The reaction was quenched by the addition of 800 μL methanol/water 5:3 mixture to each well and sealed with microplate foil. The plates were shaken at 850 rpm for 30 min prior to centrifugation at $4000 \times g$, 10 °C, for 15 min. After centrifugation, the supernatant was analyzed via LC-MS. The biotransformations were carried out once including the appropriate controls. Predicted variants (selectivity; activity) and best-performing variants (SLP; WVS) were analyzed in triplicates as individual experiments.

**Biotransformation Fl-Hal**. The cell pellets were subjected to chemical lysis using 100 μL of 25 mM HEPES buffer (pH 7.5) supplemented with 1 mg/mL lysozyme, 0.5 mg/mL polymyxin B, 0.01 mg/mL DNase and incubation for a minimum of 30 min at 20 °C on a shaking incubator at 850 rpm. Biotransformations were initiated by the addition of 100 μL of HEPES buffer (pH 7.5) containing 2 mM substrate, 0.2 mM FAD/FMN, 2 mM NADH/NADPH, 600 mM NaCl, 40 mM Glucose, and 2 μM GDH/Ec-Fre[76]. Incubation and work-up was performed in analogy to the αKGH protocol. The biotransformations with the Fl-Hal library were carried out once including the appropriate controls.

**Preparative scale biotransformation**. WelO5* SLP variant was used to prepare compound **1a** and **1b** and WelO5* WVS was used to prepare compound **1c**. For the preparation of compound **2a** the variant WelO5* VAA was used. Twenty grams of WelO5* variant cells were resuspended in 100 mL of lysis buffer (50 mM sodium phosphate, pH 8.0) containing 1 mg/mL lysozyme, 0.5 mg/mL polymyxin B, and 0.01 mg/mL DNase in a 2000 mL baffled flask. The cell suspension was shaken for a minimum of 30 min at 20 °C. Reaction was initiated by the addition of 100 mL sodium phosphate buffer (pH 8.0) containing 2 mM substrate, 220 mM α-ketoglutaric acid sodium salt, 212 mM sodium ascorbate, 1000 mM NaCl, and 2.6 mM ammonium iron(II) sulfate. The flask was incubated overnight at 20 °C on a shaking incubator at 100 rpm. About 200 mL methanol were added to the reaction mixture, and the flask was shaken vigorously. The reaction mixture was transferred to a centrifuge bottle and spun down at $4000 \times g$ for 15 min. The supernatant was transferred in a round bottom flask, and methanol was removed by a rotary evaporator. The substrate and derivatives were extracted by ethyl acetate ($2 \times 400$ mL), and the organic layer was washed with saturated NaCl solution. The organic layer was combined and dried over sodium sulfate. The solvent was removed by a rotary evaporator to yield a yellowish-brown oil.

**LC-MS analysis**. Each biotransformation sample was analyzed by LC-MS system (OpenLAB CDS 2.4). The supernatant was injected into an Agilent 1260 HPLC system equipped with a single quadrupole MSD over an Agilent Poroshell 120 EC-C18 column (2.7 μm 2.1 × 50 mm) heated at 40 °C, using water/acetonitrile 95:5 and acetonitrile containing 0.2% formic acid as solvent A and B, respectively. The following LC method was used: 0–1 min, B = 40%; 1–3 min, B = 40—100%; 3–4 min, B = 100%; 4–5 min, B = 100—40%. Fold increase in total chlorination of individual variants was normalized to a parent variant (WelO5* GAP) included as a control.

**Construction, expression, and purification of His-tagged WelO5\* variants**. His-tagged WelO5* enzyme variants (His-wt, His-GAP, His-SLP, His-VLA, and His-WVS) were created to carry out in vitro biocatalysis reactions. The mutated gene fragment encoding each variant was amplified using a primer pair 5′-GTGAGCGG ATAACAATTCCCCTCTAG-3′ (forward) and 5′-GCTTTGTTAGCAGCCGGAT CTCAG-3′ (reverse) and digested by NdeI and XhoI, which was then ligated into a pET28b(+) vector digested with the same restriction enzymes. The DNA sequence was confirmed by the DNA sequencing service provided by Microsynth AG.

Each plasmid was transformed into *E. coli* BL21(DE3) and the cells were plated on an LB agar plate containing 50 μg/mL kanamycin. A single colony of freshly transformed cells was cultured overnight in 5 mL of LB medium containing 50 μg/mL kanamycin. The culture was used to inoculate 500 mL of TB medium supplemented with 50 μg/mL kanamycin in a baffled Erlenmeyer flask. To monitor the growth of the cells $OD_{600}$ was measured and at an $OD_{600}$ of 0.6–1.0 the culture was induced with IPTG stock solution (final concentration IPTG 100 μM). Expression was carried out for 24 h at 20 °C using 120 rpm (5-cm shaking diameter). The cells were pelleted by centrifugation at $4000 \times g$, 4 °C, for 15 min, and the supernatant was discarded. The cell pellet was stored in a −20 °C freezer prior to purification. Cell pellets were resuspended in 30 mL of protein lysis buffer (50 mM Tris-HCl, pH = 7.4, 500 mM NaCl, 20 mM imidazole, 10 mM β-mercaptoethanol (β-ME), and 0.1% Tween-20) and sonicated over two rounds for 2 min with 1 s intervals on ice and then centrifuged for 30 min at $8000 \times g$ at 4 °C. The column (HisTrap™crude; 5 mL, GE Healthcare, Massachusetts, USA) was equilibrated using at least five column volumes of protein lysis buffer. The supernatant was filtered through a 0.45-μm filter and loaded onto the column. After reaching a stable UV baseline the concentration of elution buffer (50 mM Tris, pH = 7.4, 500 mM NaCl, 100 and 250 mM imidazole, and 10 mM β-ME) was raised to 100% to elute the His-tagged protein. The fractions were combined according to the UV spectra (280 nm) and the buffer was exchanged to a buffer

containing 50 mM sodium phosphate, pH 8.0. Purified protein was analyzed by SDS-PAGE to ensure its purity. The protein was concentrated using ultra centrifugal filters (Amicon® Ultra 4, cut off 10–30 kDa, Merck Millipore, MA), then flash-frozen using liquid nitrogen and stored at −80 °C. The protein concentration was determined by measuring the protein absorption via a NanoDrop spectrometer (Thermo Fisher Scientific) at 280 nm applying the estimated extinction coefficient of the protein variants (28,880 M⁻¹cm⁻¹ for His-GAP, His-SLP, His-VLA and 34,380 M⁻¹cm⁻¹ for His-WVS).

**In vitro activity assay**. In vitro activity assays were carried out in 200 μL of 50 mM sodium phosphate pH 8.0, containing 50 μM purified enzyme, 1 mM substrate, 110 mM α-ketoglutaric acid sodium salt, 106 mM sodium ascorbate, 500 mM sodium salts (NaF, NaCl, NaBr, NaI, NaN₃, and NaNO₂), and 1.0 mM ammonium iron(II) sulfate. Ninety-six well plates were sealed with breathable membranes and incubated overnight at 20 °C on a shaking incubator at 850 rpm. The reaction mixtures were quenched with an 800 μL methanol/water mixture (62% methanol). The plate was sealed with microplate foil and shaken at 850 rpm for 30 min prior to centrifugation at 4000 × g, 10 °C, for 15 min. Each biotransformation sample was analyzed by LC-MS system using selected ion monitoring (SIM).

Formation of **1a** was quantified through a calibration curve (Supplementary Fig. 11) prepared from known concentrations of the product isolated by the preparative scale biotransformation. As internal standard (ISTD) 0.4 mg/L soraphen C was used.

To determine $k_{cat}$ (chlorination of soraphen A), assays were carried out in an identical manner as the assay described above except that reactions were performed at different substrate concentrations (Supplementary Table 6) and with the addition of 3.8% dimethylformamide (μL/μL). At indicated time points (1, 2, 3, 4, and 5 min), 20 μL of reaction mixture was transferred into 980 μL of methanol/water mixture (methanol:water = 1:1 + 0.4 mg/L soraphen C) to quench the reaction. The product formation was monitored by LC-MS and was plotted over time, which was then fitted by linear regression using Microsoft Excel. The observed initial rates were fitted to a substrate inhibition model (Supplementary Eq. 1 and Supplementary Fig. 12) using GraphPad Prism 8.4.0 (nonlinear regression) with the following restraints: $K_m > 0$, $K_i > K_m$. TTN was determined at a substrate concentration of 60 μM and the reaction was quenched at stable product concentration using the same procedure as above. The following enzyme variant concentrations were used: GAP = 5 μM, SLP = 0.5 μM, and VLA = 0.1 and 0.5 μM.

**Ligand docking and homology modeling of WelO5* variants**. Models of the wild type WelO5* and the WelO5* variants were created using the SWISS-MODEL[56] online server with default parameters. The crystal structure of wild type WelO5 (PDB ID: 5J4R) served as a template for the homology modeling. The docking process was performed using default parameters of Chimera AutoDock Vina[77] and the region of interest was set to default, as this docking is flexible. Each docking result was visually inspected using PyMOL 2.4.1 software.

**Machine learning**. The label vector was defined as activity or selectivity. The activity label ($A$) was calculated using the formula $A = tot.$ Cl conversion WelO5* mutant / tot. Cl conversion WelO5* GAP whereas tot. Cl conversion = $(SIM_{1a} + SIM_{1b}) / (SIM_{1a} + SIM_{1b} + SIM_{1c} + SIM_1)$. The selectivity label ($S$) was calculated using the formula $S = (SIM_{1a} – SIM_{1b}) / (SIM_{1a} + SIM_{1b})$. Amino acids were represented as a 17-dimensional vector, which was obtained by concatenating the five-dimensional T-scale descriptor[63] and additional information about amino acid characteristics[64]. We then defined the feature vector of a sequence by joining the vector representation of its individual amino acids at sites V81X, A88X, I161X, and aggregated them into the 504 × 51-dimensional training matrix. This was used to train a machine learning model, based on the Algorithm 2.1 of Gaussian Processes for Machine Learning (GPML) by Rasmussen and Williams[78], implemented in the scikit-learn python module. We took a similar approach for predictions of selectivity; however, we excluded variants below a peak area threshold and relied on the random forest implementation in scikit-learn for predictions, using the same input features as for activity. To avoid overfitting and to better gauge the generalizability of our model, we cross-validated over ten splits, and model performance was evaluated on the coefficient of determination ($R^2$), a standard metric for regression problems, achieving an out of fold score of 0.745/0.31 for activity/selectivity respectively (compare predicted vs. measured Supplementary Fig. 13). Inference occurred on the remaining sequence space, which was preprocessed exactly like the training data, at every fold during cross-validation. The code, data, and supplementary information, such as amino acid encodings, can be accessed at: [https://github.com/ccbiozhaw/MLevo].

**Reporting summary**. Further information on research design is available in the Nature Research Reporting Summary linked to this article.

## Data availability

Source data are provided with this paper. WelO5 crystal structure used as template for SWISS-MODEL homology modeling can be accessed via PDB ID: 5J4R. The authors declare that all the data supporting the findings of this work are available within the article and its Supplementary Information and the provided Source Data. Source data are provided with this paper.

## Code availability

Training data and scripts used to predict enzyme function are available at https://github.com/ccbiozhaw/MLevo, https://doi.org/10.5281/zenodo.5665270

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

## Acknowledgements

We thank Myriam Baalouch for her help with the synthesis and analytics of the soraphen compounds, Dirk Balmer for the biological testing of the compounds, Dianne Irwin for the purification of the chlorinated products, Federico Dapiaggi for his help with the docking experiments and helpful discussions, and Leonard Hagmann for his help with the NMR analysis of the products. Furthermore, we thank An Vandemeulebroucke for the helpful discussion of the kinetic results. This work was supported by the Swiss State Secretariat for Education, Research and Innovation (Federal project contributions 2017–2020, P-14: Innovation in Biocatalysis) and was created as part of NCCR Catalysis, a National Centre of Competence in Research funded by the Swiss National Science Foundation (Grant number 180544).

## Author contributions

J.B., A.L., C.L.C., O.L. and R.B. designed the research. J.B. carried out most of the experiments and performed the docking analysis. S.H.M. and M.V. constructed the combinatorial enzyme library. D.P. carried out the machine learning predictions. O.A. synthesized the soraphen analogs. C.L.C. analyzed the NMR structures. J.B., N.J.T., U.T.B., C.L.C., O.L. and R.B. discussed the results and wrote the manuscript.

## Competing interests

Author O.A. is employed by Idorsia Pharmaceuticals Ltd and authors C.C., A.L. and O.L. are employees of Syngenta Crop Protection AG. The remaining authors declare no competing interests.

## Additional information

**Peer review information** *Nature Communications* thanks Kinshuk Raj Srivastava and the other anonymous reviewer(s) for their contribution to the peer review this work. Peer reviewer reports are available.

