## [Peer Review File · Nature Communications]

Reviewers' Comments:

Reviewer #2 (Remarks to the Author)

I have read through the revised manuscript and the authors' responses to my concerns. To the best of my knowledge, all of my concerns have been addressed and the authors have included the additional data that I requested in my initial review. I believe that this manuscript is appropriate for publication in Nature Communications. Just one correction: even though the authors noted that the word "inactivated" (line 18) has been corrected to "unactivated" in their response letter, this correction was not made in the revised manuscript.

Reviewer #1 (Remarks to the Author):

The manuscript submitted by Buchler, J. *et. al.* reports late-stage halogenation of soraphen family of macrolides-based natural products through machine learning assisted enzyme engineering strategy. Using a new enzyme engineering strategy, they could evolve WelO5 to accept and halogenate non-native substrate i.e. soraphen macrolides, and measured the antifungal activity, however, variants showed lower activity compared to soraphen. This study provides a complete picture which includes ML-based prediction and biocatalysis through engineered variants, and activity testing of the diverse halogenated product.

In the previous round of review, I had many concerns and I found the current version of the manuscript tried addressing most of my concerns related to data interpretation, clarity improving, and minor corrections in NMR data too. In addition to addressing all the concerns, the authors have made the required changes in the manuscript for better readability.

The current version of the manuscript is in better shape compared to its previous version. Based on the newly added data related to activity and rationale of regioselectivity for SP3 carbon halogenation, adjustments in NMR data, as well as text edits for improved clarity, **I support the manuscript for publication in Nature Communication.**

We are grateful to Reviewer 1 for reviewing our manuscript and his/her support of the manuscript for publication in Nature Communications.

I have the following concerns and suggestions:

1. The author tested ~500 unique variants from the library and was able to observe several variants with improved chlorination activity. I couldn't find the explanation for the molecular basis

of fold enhancement of variants, for instance, SLP and VLA compared to GAP. As per K_m (table 1) the affinity for ligands are similar only changes were observed in K_{cat} . I am curious about the % conversion of soraphen by these mutants as well. Interestingly, added new data demonstrate that SLP has a more relaxed substrate scope compared to the other two variants tested and it could be possible that VLA may also have relaxed substrate scope and can accept monochlorinated substrates. Can the author explain the observation of fold enhancement and relaxed substrate scope of the SLP and VLA variant?

In response to Reviewer 1's comments, we additionally carried out *in vitro* substrate scope experiments for VLA, an improved variant, which we had previously not investigated in this context. As seen for SLP, variant VLA shows a more relaxed substrate scope than detectable for variants GAP or WVS leading to 1.9 % doubly chlorinated and 5.8 % chlorinated and hydroxylated product when presented with substrate 1b. We included these findings in the main text and in Figure S7.

Going forward, we then carried out the comparative % conversion experiments using freshly purified variants GAP, SLP and VLA with soraphen A as the substrate (new Figure S12). Considering identical enzyme and substrate concentrations, variant GAP showed a very poor total conversion of below 1 %, whereas SLP generated on average 30 % of product 1b and VLA outperformed both other variants reaching approximately 50 % conversion. We agree with reviewer 1 that the reason is likely not to be found in the K_m (which is similar for all variants) but could be explained by the improving total turnover numbers exhibited by these enzymes (GAP = 0.3; SLP = 30; VLA = 92). In conclusion, it is conceivable that all WelO5*-derived halogenases can principally accept the mono-chlorinated soraphens, albeit so poorly that this becomes detectable only under optimal reaction conditions and only for the most active, engineered variants. We have reflected these considerations in the main text by rewriting the corresponding paragraph.

To explore why we observe a difference in fold enhancement with our engineered variants, we followed up by creating AlphaFold models of the WelO5* wildtype enzyme as well as of variants GAP, SLP and VLA. In this context, we investigated the substrate access tunnel using the computational tool Caver Web 1.0, which allowed us to analyze the bottleneck radii of the substrate entrance tunnels. This investigation highlighted that the employed enzyme engineering approach led to a widening of the access tunnel from 2.1 Å (wildtype enzyme) to 3.2 Å (variant GAP), 2.6 Å (variant SLP) and 2.4 Å (variant VLA) (new Figure S13). The resulting improved access to the active site might explain why the wildtype enzyme cannot convert the macrolide soraphen A at all, while GAP, SLP and VLA accept the bulky substrate.

Overall, we believe that the observed fold enhancement can at least in part be explained by an improved substrate access to the active site. However, additional factors, which are currently more difficult to capture, such as an improved relative placement of the substrate in relation to the Fe(IV)=O species likely also play a role in the overall reaction outcome of the different variants and govern enzyme characteristics such as total turnover number, fold enhancement and regioselectivity.

2. Now based on other reviewer's comments they have provided required data (MLbased algorithm and training data sets) on Github. I have gone through their training datasets and realized that they have included only activity and selectivity and amino acid property only to train their ML model. Although the data size is less, all experimental data are their own, however, I understand the limitation of data availability for soraphen which is a non-native substrate for WelO5. I feel that the inclusion of data sets regarding protein and ligands conformational space, and energetics of binding will add more precision in tuning their ML model for better activity and selectivity prediction.

We thank the reviewer for his constructive suggestions and will certainly endeavor to further tune our ML model in this way for future investigations. To reflect these thoughts, we have adapted the summary and outlook section accordingly.

This was also a reason I recommended them to perform the MD simulation to learn about the conformational landscape of WelO5 and soraphen. Although soraphen being macrolide structure will have limited conformational space, still it can provide a better understanding for possible conformation and can add value in determining the preferred binding in the active pocket and regioselectivity. Simulation of PKS based macrolide structure in P450 enzyme is very routinely done and published, therefore, I completely disagree with the author's justification (for comment 2) that "Regarding MD simulation, the result we obtained were not interpretable (substrate exited the active pocket), highlighting how difficult these calculations are when dealing with a bulky and flexible substrate and an enzyme that has been modified compared to the native one". The observation of substrate exited active pocket could be just a simulation error and parameters were not optimized properly. I am curious to see the Rg plot of simulation for substrate and a-KG.

We thank the reviewer for his constructive suggestions.

MD calculations were re-performed as the initial MD simulations indeed showed an artefact with the substrate exiting the active pocket. Subsequent calculations indicated that the substrate stayed in the active site, however, the generated results did not bring any satisfactory insights that could explain the observed selectivity or guide the protein evolution process. Considering that the presented experimental results are unambiguous (NMR structure elucidation), we believe that publishing the MD simulations would be confusing to the reader and would not add any value to the manuscript.

Since this is a non-native substrate, therefore, the binding could be poor too that could also lead to dissociation of soraphen from the WelO5 pocket. If the poor binding of non-native substrate (soraphen) could be the reason then I am curious to see the % conversion of soraphen by WelO5 and a few of their best variant which may provide a clue.

Apart from carrying out % conversion experiments (see reply to point 1), in which we observed good % conversion for VLA and SLP, we additionally set out to answer the question of ligand binding using enzyme homology models generated by AlphaFold docking soraphen A (1), mono-chlorinated soraphen A (1a) and the native substrate 12-epi-fischerindole U into the enzyme active site. Comparing the average docking scores of all valid solutions (e.g., solutions for which soraphen A docked into the active site), we found that all variants showed a similar average docking score for all investigated substrates, independent of them being native or un-natural substrates.

Considering the observed % conversions, which seem to be governed by the total turnover numbers of the respective variants, we currently do not think that poor binding of soraphen is the main limiting factor in the biocatalysis reactions.

3. MD data can also provide details of other key positions interacting with soraphens or helping in reorienting which can also be further considered for mutational hotspot. Based on previous studies (ref 43 and 23) performed on a different substrate and their docking studies with soraphen authors selected 3 key positions for randomization and have trained their model to predict selectivity and activity. I believe that identification of additional residues in the binding pocket for mutagenesis will be further helpful in the altered binding of soraphen in the WelO5 active pocket and that could lead to altered regioselectivity of halogenation. However, I understand that increment of additional position will expand the vector size and with limited availability of training data it can lead to many solutions and therefore erroneous prediction. The learning and the acquired data from this study which combined with additional data on the conformational landscape of protein and ligand can further help in the iterative round of learning and tuning their ML model for better and precise prediction of activity and altered selectivity. Maybe through this approach, they may be able to generate better mutants in the future with improved binding and altered regioselectivity which can enhance the bioactivity of soraphen macrolides. So far they

could not reach to their goal of developing better soraphen derivatives with the improved anti-fungal activity which is required for commercial level exploration of soraphen analog in agroindustry.

We thank Reviewer 1 for his valuable advice. While we are confident that further optimization of our WelO5* variants will be possible, the goal of the current work was to explore new avenues which would allow to create interesting molecules as part of agrochemical discovery efforts. Toward this goal, we have shown that halogenase WelO5* can be rapidly engineered, thanks to computational modelling including the application of machine learning algorithms, to deliver late-stage functionalized macrolides in quantities which are sufficient for agrochemical testing. This approach can now be applied (and further fine-tuned) by us and others in future late-stage functionalization projects. To reflect Reviewer 1's consideration of using MD calculations and larger libraries in future evolution campaigns, we expanded the summary and outlook paragraph accordingly.

Reviewer #2 (Remarks to the Author):

I have read through the revised manuscript and the authors' responses to my concerns. To the best of my knowledge, all of my concerns have been addressed and the authors have included the additional data that I requested in my initial review. I believe that this manuscript is appropriate for publication in *Nature Communications*. Just one correction: even though the authors noted that the word "inactivated" (line 18) has been corrected to "unactivated" in their response letter, this correction was not made in the revised manuscript.

We are grateful to Reviewer 2 for reviewing our manuscript and his/her valuable feedback and support. We apologize for the oversight regarding the above-mentioned word "inactivated". We have made the correction to "unactivated".

We hope that our revisions performed in the main text and the Supporting Information as well as the clarifications described above to the constructive and helpful comments made by the two reviewers will now lead to acceptance of our manuscript for publication in *Nature Communications* and very much look forward to your reply.

With best regards,

Reviewers' Comments:

Reviewer #1:

Remarks to the Author:

I have gone through the revised version of the manuscript and the response letter that the authors submitted. I believe that all my concerns have been addressed in the revised version of the manuscript and the authors have performed relevant experiments to justify their comments. Accordingly, they have incorporated the discussion in the main manuscript as well. The current version of the manuscript, I believe is mature enough and is suitable for publication in Nature Communication.